# Behavior of Assembled Promyelocytic Leukemia Nuclear Bodies upon Asymmetric Division in Mouse Oocytes

**DOI:** 10.3390/ijms25168656

**Published:** 2024-08-08

**Authors:** Osamu Udagawa, Ayaka Kato-Udagawa, Seishiro Hirano

**Affiliations:** Environmental Risk and Health Research Division, National Institute for Environmental Studies, Tsukuba 305-8506, Japan

**Keywords:** nuclear bodies (NBs), oocyte maturation, fertilization, membrane-less organelle, stress response

## Abstract

Promyelocytic leukemia (PML) nuclear bodies (PML-NBs) are core–shell-type membrane-less organelles typically found in the nucleus of mammalian somatic cells but are absent in mouse oocytes. Here, we deliberately induced the assembly of PML-NBs by injecting mRNA encoding human PML protein (hPML VI -sfGFP) into oocytes and investigated their impact on fertilization in which oocyte/embryos undergo multiple types of stresses. Following nuclear membrane breakdown, preassembled hPML VI -sfGFP mRNA-derived PML-NBs (hmdPML-NBs) persisted in the cytoplasm of oocytes, forming less-soluble debris, particularly under stress. Parthenogenetic embryos that successfully formed pronuclei were capable of removing preassembled hmdPML-NBs from the cytoplasm while forming new hmdPML-NBs in the pronucleus. These observations highlight the beneficial aspect of the PML-NB-free nucleoplasmic environment and suggest that the ability to eliminate unnecessary materials in the cytoplasm of metaphase oocytes serves as a potential indicator of the oocyte quality.

## 1. Introduction

In the nucleus of a typical mammalian cell, there are structures called promyelocytic leukemia nuclear bodies (PML-NBs) with a diameter of about 0.2–1 μm. Microscopically, PML-NBs are often doughnut-shaped, and PML protein constitutes its shell. The inside of the shell is thought to contain more than 100 proteins (clients) [1]. PML-NBs are known to be involved in various reactions, including DNA damage response [2,3] and viral infection [4,5], depending on the clients. PML-deficient mice show immune system phenotypes such as leukopenia and reduced innate immunity [6,7], but no reproductive phenotypes have been reported, suggesting that PML plays an auxiliary role in reproduction. Given active fluxes of non-membranous organelles in the oocyte nucleus during growth, it remains curious as well as unclear why oocytes are devoid of specific membrane-less organelles such as Cajal bodies [8] and PML-NBs [9].

Small Ubiquitin-like Modifier (SUMO) conjugation is known as one of the post-translational modifications of proteins [10,11]. In typical somatic cells, SUMO is often well concentrated in PML-NBs when observed under a fluorescent microscope, and it can be seen that PML-NBs act as a reservoir of SUMO [12]. In mouse oocytes, we previously found that SUMO is recruited to large droplets with a diameter of over 5 μm when oocytes are exposed to stresses such as heat shock or insufficient proteolysis. In the presence of artificially assembled PML-NBs, SUMO was sequestered by the preassembled PML-NBs even under stress conditions, altering the stress-coping efficiency of SUMO [9].

From another angle, we wanted to approach the reason why oocytes do not retain PML protein in the form of nuclear bodies by focusing on the asymmetric division of oocytes which is specific to germ cells. To be competent to fertilize, oocytes pause their transcription, undergo breakdown of a germinal vesicle (GV) which is surrounded by nuclear membrane (GVBD), and divide asymmetrically, releasing the smaller daughter cell as a first polar body (which is no longer necessary in the subsequent growth process) [13]. In each round of repetitive mitotic division of somatic cells after the nuclear membrane breakdown, PML-NBs contact each other to gather in the cytoplasm, called mitotic accumulations of PML protein (MAPPs). Bypassing a proteasomal degradation, parts of PML proteins in MAPPs are recycled to form new PML-NBs in the nuclei of daughter cells [14]. In this study, we attempt to clarify how artificially assembled PML-NBs are expelled into the cytoplasm after GVBD and behave during pronucleus reformation at fertilization. Our previous methods for the artificial assembly of PML-NBs in oocytes involved the injection of plasmid DNA into the nuclei of young oocytes that are highly transcriptionally active [9]. However, this method was not suitable for the analysis of transcriptionally silent, mature oocytes. In this study, we performed further analysis by injecting mRNA synthesized by in vitro transcription into oocytes and subsequent developmental stages.

## 2. Results

We previously reported that endogenous PML-NBs are gradually formed from approximately the four-cell stage onwards (Figure 1A). To clarify how PML-NBs artificially assembled in transcriptionally silent germinal vesicle (GV) oocytes behave during, and after meiotic maturation, and whether there are any disadvantages, we first designed plasmids to synthesize mRNA for human PML protein translation. Injection of 25 ng/μL of EGFP-hPML VI mRNA resulted in uniform distribution in the nucleoplasm (Appendix A).

In contrast, the injection of 150 ng/μL resulted in the formation of nuclear bodies, despite being somewhat faint (Appendix A). Next, we attempted to attach Superfolder GFP (sfGFP) to the C-terminus of hPML VI, hoping to attain efficient nuclear body formation. sfGFP corresponds to the S30R, Y39N, N105T, Y145F, I171V, and A206V mutants of GFP and is known to be characterized by a rapid folding [15]. The injection of 25 ng/μL of hPML VI -sfGFP mRNA into the cytoplasm of GV oocytes resulted in the formation of evident nuclear bodies (Figure 1B). Hereafter, we call these hPML VI -sfGFP mRNA-derived PML-NBs hmdPML-NBs. Given that exogenously expressed human PML is well capable of sequestering endogenous mouse PML clients (Appendix A, see also [9]), potential PML clients in the mouse oocytes have an ability to constitute PML-NBs if a certain unknown condition is fulfilled. While the nucleation of hmdPML-NBs in the nucleoplasm would likely follow a manner similar to other exogenous expression methods (e.g., CHO-K1 cells stably expressing GFP-hPML VI, see Appendix A), whether or not exogenous PML protein “purely” self-aggregates into hmdPML-NBs will require in vitro droplet experiments.

The integrity of the nuclear envelope can be maintained with the phosphodiesterase inhibitor IBMX (3-Isobutyl-1-methylxanthine), allowing GV oocytes enough time to translate the injected mRNA. To proceed to further stages, IBMX was removed from the culture media (in vitro maturation: IVM). At the stage of GVBD (4 h after removing IBMX), hmdPML-NBs were distributed in a location independent of the condensed DNA (Figure 2A). At 20 h after removing IBMX, spindles positive for α-tubulin antibody were formed not only on the polar body side but also within the oocyte cytoplasm. hmdPML-NBs were distributed in the cytoplasm independently of the location of these spindles, and they were not distributed around the condensed DNA (Figure 2B,C).

Oocytes that have released polar bodies are basically fertilizable with sperm. To overcome the hardening of zona due to prolonged in vitro culture, we induced pronuclear reformation by parthenogenetic stimulation using SrCl_2_ and cytochalasin under Ca^2+^-free conditions. Expelled hmdPML-NBs preassembled at the GV stage often remained in the cytoplasm of arrested MII oocytes (Figure 3A). The number of pronuclei successfully formed in parthenogenetic embryos was one or two. In addition to the faint nucleoplasmic distribution of hPML VI -sfGFP, hmdPML-NBs were clearly visible as NBs at an average of 4.20 ± 0.57 per pronucleus. In such parthenogenetic embryos, expelled hmdPML-NBs were no longer visible in the cytoplasm (Figure 3B,C). To understand the newly formed pronuclar hmdPML-NBs including their origin in detail, we injected hPML VI -sfGFP mRNA into one-cell embryos after insemination with sperm. As a result, an average of about four hmdPML-NBs were similarly assembled in the nuclei of two-cell embryos. Notably, the larger hmdPML-NBs formed were prone to be toric (Figure 3D). These results likely suggest the clearance of preassembled hmdPML-NBs in the successfully fertilized embryos. Also, a limited amount of mRNA remaining in the cytoplasm would be translated to form relatively smaller numbers of pronuclear hmdPML-NBs. It has been reported that SUMOylation via the environment of PML-NBs is important for the activity of proteins involved in cell fate determination immediately after fertilization (KAP1: transposon erasure, DPPA2: 2-cell-ness) [16,17,18]. KAP1 was sequestered by the assembled hmdPML-NBs in the nuclei of two-cell embryos but only to a limited extent (Appendix A). In four-cell embryos in which hmdPML-NBs were assembled after fertilization, an increase in DPPA2 expression was observed, as in the control group (Appendix A). These results suggest that assembled hmdPML-NBs after fertilization do not show an overt defect on the progression of development.

Considering the stress-responsive property of PML protein [19,20], we further analyzed the persistence of preassembled hmdPML-NBs in the cytoplasm of metaphase oocytes that are preparing for the pronuclear membrane reformation in the presence of arsenite. First, we confirmed that arsenic was taken up by oocytes by monitoring the decay time from arsenic administration (Figure 4A). Based on the observation that spindle formation was inhibited by 10 μM arsenite treatment but maintained at 3 μM (Figure 4B,C), IVM was performed in the presence or absence of 3 μM arsenite for 21 h. As a result, a greater number of preassembled hmdPML-NBs persisted in the cytoplasm than in the absence of arsenite (Figure 4D,E). Observation of GV-arrested oocytes that did not undergo GVBD is useful for inferring the efficiency of microinjection as well as the appearance during the GV stage of oocytes that successfully underwent GVBD. After 21 h of culture with arsenite, hPML VI -sfGFP was clearly accumulated as hmdPML-NBs in the nuclei of GV-arrested oocytes and was less distributed in the nucleoplasm than in the absence of arsenite (Figure 4F), indicative of the insolubilization in the nucleoplasm. It is generally believed that in the presence of arsenite, PML proteins in the nucleus are modified by SUMO distributed in the nucleus which mediates their degradation via SUMO-dependent Ubiquitin E3 ligase system [1]. When the culture was performed for 40 h, hmdPML-NBs were still found to remain in the cytoplasm even in the presence of arsenite (Appendix A). On the other hand, in GV-arrest oocytes treated with arsenite, the disappearance of preassembled hmdPML-NBs was observed in many oocytes, with a few oocytes in which strongly aggregated hmdPML-NBs remained (Appendix A). This suggests that the degradation system that potentially works well in the nucleus does not work sufficiently on preassembled hmdPML-NBs that have been dispersed in the cytoplasm after GVBD. Together, if oocytes retain PML protein in the nucleus in the form of NBs like typical somatic cells, it may become nuisance debris for pronuclear formation in the cytoplasm which could be enhanced under stress.

## 3. Discussion

PML-NBs are thought to be non-membrane organelles that consist of a proteinous scaffold (PML protein) and clients. In the ovaries and fallopian tubes, oocytes/embryos undergo various types of stresses (Figure 5). Given that the impaired dynamics of non-membrane organelles with irreversible aggregate formation due to excessive or chronic stress are related to pathological conditions [21], it would be important to analyze the optimal timing of the emergence, abundance, and distribution of non-membrane organelles in the light of the physiological environment for the development of oocytes.

Reasonably, as an example of another organelle, energetic mitochondria are known to preferentially persist in the cytosol rather than be discarded to the polar body [22]. Thus, initially, we had expected that preassembled hmdPML-NBs would be discarded to the polar body because they were likely unnecessary. Actually, they persisted in the cytoplasm of oocytes. Non-membrane organelles generally have an advantage: a small change (e.g., stress) can trigger a large change in biopolymers [23] that spreads throughout the cell (e.g., from nucleoplasmic distribution to NBs) saving the cost of synthesis and degradation. In oocytes, however, the stress-responsive property of PML protein may worsen the situation. Observation of ovulated oocytes of *hPML VI -sfGFP* Transgenic mice or injection experiments of non-invasive stimuli (light or something similar)-responsive biopolymers would probably tell us details on the responses of hmdPML-NBs to more physiological stress, as well as make it clear whether the debris per se affects the preparation of fertilization.

PML protein is not present in the form of NBs at least in mouse oocytes. We previously reported that endogenous PML protein (detected with clone 36.1–104 LOT#2477976) appears to be attached to condensed DNA after GVBD [9]. According to recent research, the peri-chromosomal accumulation is one of the different types of phase separation from the NB formation [24,25]. The peri-chromosomal localization of PML protein could not be observed with newly available PML monoclonal antibody (LOT#3989079), while we have confirmed the appearance of endogenous PML-NBs in preimplantation embryos. Further verification using PML KO oocytes is awaited.

As for the origin of newly formed hmdPML-NBs in the pronucleus, there was no difference in the number of NBs per pronucleus between the embryos with one and two pronuclei. Also, the injection of hPML VI -sfGFP mRNA into ovulated MII oocytes (with no preassembly of hmdPML-NBs) resulted in a similar number of NBs per pronucleus (see details in the Open peer review). Therefore, it was not likely that the preassembled hmdPML-NBs in the cytoplasm were directly reused as new NBs. Rather, the reformation of the (pronuclear) nucleoplasmic environment made it possible for the reappearance of hmdPML-NBs. Since the recycling of PML-NBs or other non-membrane nuclear organelles for daughter cells is reported during mitosis [14,26], the involvement and its contribution of recycling of the PML protein warrants further investigation.

Finally, given that the embryos that successfully formed pronuclei were able to erase the preassembled hmdPML-NBs in the cytoplasm while forming new hmdPML-NBs in the pronucleus, it is interpretable that the clearance of debris in the cytoplasm is advantageous for embryo development. Although the detailed mechanism for hmdPML-NBs’ degradation is unknown and further analysis is required, this study has also provided a perspective on the clearance ability in the cytoplasm as an oocyte quality.

## 4. Materials and Methods

### 4.1. Chemicals, Reagents, and Antibodies

Sodium arsenite (NaAsO_2_), Triton X-100, cytochalasin D, strontium chloride (SrCl_2_), sodium pyruvate, and 3-isobutyl-1-methylxanthine (IBMX) were purchased from Sigma (St. Louis, MO, USA). Paraformaldehyde (PFA), ethanol, and bovine serum albumin (BSA) were purchased from WAKO (Osaka, Japan). Pregnant mare serum gonadotropin (PMSG) and human chorionic gonadotropin (hCG) were purchased from ASKA Pharmaceutical (Tokyo, Japan). Modified human tubular fluid (mHTF) medium and K-modified simplex optimized medium supplemented with amino acids (KSOM-AA) were purchased from Kyudo (Saga, Japan). α-minimum essential media (MEM) medium, penicillin/streptomycin, and 4-(2-hydroxyethyl)-1-piperazineethanesulfonic acid (HEPES) were purchased from Gibco/Thermo Fisher Scientific (Grand Island, NY, USA). Recombinant human epidermal growth factor (EGF) was purchased from Peprotech/Thermo Fisher Scientific (Cranbury, NJ, USA). Mineral oil was purchased from Nacalai Tesque (Kyoto, Japan). Hoechst dye was purchased from Dojin Chemical (Kumamoto, Japan). Fetal bovine serum (FBS) was purchased from Biowest (Nuaille, France). The following antibodies were used in this study: anti-human PML (sc-966: Santa Cruz Biotechnology, Dallas, TX, USA), anti-human PML (A301-167A: Bethyl, Montgomery, TX, USA), anti-DPPA2 (MAB4356) (Millipore, Burlington, MA, USA), anti-α-Tubulin (CP06: Sigma), anti-KAP1 (15202-1-AP: Proteintech, Rosemont, IL, USA), and Alexa 488- or 594-conjugated secondary antibodies (Molecular Probes/Thermo Fisher Scientific, Waltham, MA, USA).

### 4.2. Collection and Culture of Oocytes, Zygotes, and Embryos

All animal procedures and protocols were in accordance with the Guidelines for the Care and Use of Laboratory Animals and were approved by the Animal Care and Use Committee of the National Institute for Environmental Studies. C57BL/6J mice were purchased from CLEA-Japan (Kawasaki, Japan). The animals were housed under a 12 h/12 h light/dark cycle with free access to food and water. Unless otherwise mentioned, oocytes were cultured in medium for in vitro maturation (basal medium: α-MEM medium supplemented with penicillin/streptomycin (100 units/mL and 100 μg/mL, respectively) and heat-inactivated FBS (10%)). The collection of fully grown germinal vesicle (GV) oocytes was conducted as previously described [9]. Briefly, follicles on the ovarian surface were mechanically ruptured with a pair of forceps to isolate GV oocytes in basal medium supplemented with IBMX (a phosphodiesterase inhibitor that can inhibit/control meiotic resumption). For observation of metaphase stage oocytes, IBMX was washed out with basal medium to allow meiotic resumption. Development of oocytes was further verified by the extrusion of the first polar body (Pb1; a marker of MII-stage oocytes that have the potential to be fertilized with sperm) after an additional incubation time as indicated in each experiment. To obtain preimplantation embryos, female mice were initially primed by intraperitoneal injection of PMSG followed 48 h later by injection with hCG. Superovulated oocytes were collected from the oviducts of euthanized mice by gently teasing apart the ampulla with a 21-gauge needle (TERUMO, Tokyo, Japan) to release cumulus-oocyte complexes in mHTF medium; these oocytes then were inseminated with pre-capacitated sperm. Cultures were performed in a drop of medium (30–80 µL) under a mineral oil overlay at 37 °C in a humidified atmosphere of 5% CO_2_ (APM-30D; ASTEC, Fukuoka, Japan).

### 4.3. RNA Microinjection

By customization (Vector builder, Yokohama, Japan), mutated green fluorescent protein (EGFP or sfGFP)-encoding regions were placed upstream or downstream of the coding sequence of *Pml VI* (human PML transcript variant 5, NM_033244) in the plasmids for in vitro transcription. Each of resulting mRNAs driven by T7 promotor was designated as EGFP-hPML VI or hPML VI -sfGFP mRNA, respectively. Each mRNA was diluted with RNase free water (Jena Bioscience, Jena, Germany) to the concentration as indicated. Each solution was loaded into an injection pipette (TIP-DNA [LIC-OD1], NAKA Medical, Tokyo, Japan), and the pipettes were placed in an IM-9B microinjector (Narishige, Tokyo, Japan). MN-4 and MMO-202ND manipulators (Narishige) were adapted to an IX70 inverted microscope (Olympus, Tokyo, Japan) via NO-PIX-4-P (Narishige). mRNA solutions were injected into each oocyte/embryo cytoplasm until adequate swelling of the plasma membrane was observed.

To obtain parthenogenetic embryos with PML-NBs, GV-oocytes were first collected from ovaries that were primed in advance with PMSG for 48 h. The oocytes microinjected with hPML VI -sfGFP mRNA were kept in GV-stage by 50 µM IBMX for 7–8.5 h. In vitro maturation was carried out and additionally supplemented with 25 µg/mL sodium pyruvate, 0.1 IU/mL PMSG, 1.2 IU/mL hCG, and 4 ng/mL EGF. Matured oocytes were further treated with 5 mM SrCl_2_ and 5 µg/mL cytochalasin in Ca^2+^-free M16 medium for up to 6 h to activate and diploidize, respectively. After the activation, oocytes were further cultured in KSOM-AA. To obtain preimplantation embryos with PML-NBs, after 7 h of insemination, embryos were washed out of sperm and microinjected with mRNA as described. Microinjected embryos were further cultured until the indicated embryonic stages in KSOM-AA.

### 4.4. Immunofluorescent Staining

Except for the live imaging of GV oocytes in Figure 1B and Appendix A, oocytes, zygotes, and embryos at each indicated stage were fixed with 4% phosphate-buffered PFA (pH 7.0–7.4) at room temperature. Cells were permeabilized with 0.5% Triton X-100 in PBS, blocked with 5% BSA-PBS, and then stained with the indicated primary antibodies in 1% BSA-PBS. Cells were visualized with Alexa-conjugated secondary antibodies; Hoechst dye was included to stain DNA. Cells were placed in small drops (4 µL each) of 1% BSA-PBS, covered with mineral oil, in a glass-bottomed 35-mm petri dish (AGC techno glass, Shizuoka, Japan). Images were captured by confocal microscopy (Leica TCS-SP5; Leica, Solms, Germany).

### 4.5. Measurement of Arsenic Content by Inductively Coupled Plasma Mass Spectrometry (ICP-MS)

Oocytes (similarly collected as described above but on ice and washed thoroughly) or ovaries (and liver as a positive control) were collected on ice after indicated time courses after the intraperitoneal injection of saline (negative control) or 8 mg/kg bw sodium arsenite. Concentration of arsenic in ovary or oocyte was determined by ICP-MS (7500 cx, Agilent Tech., Santa Clara, CA, USA) after wet-ashing each biological samples with 600 µL of HNO_3_ and 200 µL of 30% H_2_O_2_ (HNO_3_:H_2_O_2_ = 3:1) under refluxing condition at 135 °C for 2 days. The samples were diluted with ultra-pure Milli-Q (MQ) water at a final volume of 5 mL for the ovary and 1 mL for the oocytes. The number of GV-oocytes used for detection was as follows: 1 h: 213–283 oocytes (7–9 mice); 4.5 h: 144.5–154 oocytes (7 mice each, a half of the volume analyzed); 24 h: 566 oocytes (13 mice). Total arsenic concentrations were measured in the normal mode (7500 cx; *m*/*z* 75: ^75^As^+^). Bovine liver (NIST, Gaithersburg, MD, USA) was used as the certified reference material (CRM). The analytical method was validated by measuring the arsenic concentration in the CRM. The estimation process for relative volumes of the ovary and oocyte was as follows. The ovary and oocyte were regarded as an oval sphere and a sphere, respectively. Ovary: major axis (M): 1918 μm; minor axis (m): 1095 μm; V = 4/3 π × 1/2 M × (1/2 m)^2^ = 4/3π × 959 × 547 × 547 ≅ 1.2 mm^3^. Oocyte: diameter (D): 90 μm; V = 4/3π × (1/2 D)^3^ = 4/3π × 45 × 45 × 45 ≅ 0.0004 mm^3^. V(Ovary)/V(oocyte) = 1.2/0.0004 = approx. 3000.

### 4.6. Data Analysis

Data were analyzed by *t*-test. The comparison between groups was considered to be significantly different when *p* < 0.05. Where indicated, data are presented as means with the standard error of the mean (SEM).

## Figures and Tables

**Figure 1 ijms-25-08656-f001:**
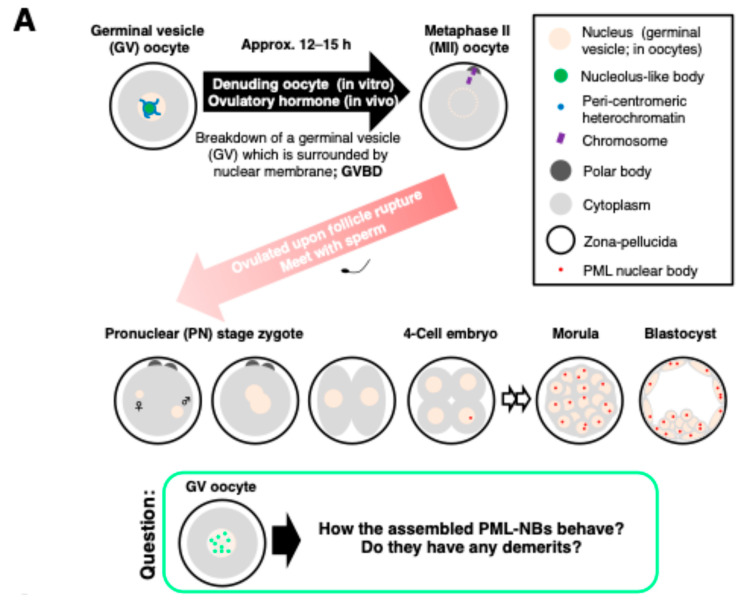
Successful assembly of PML-NBs in the nucleus of GV oocytes. (**A**) Oocytes headed for fertilization with sperm upon ovulation as the metaphase II (MII)-stage oocyte. Alternatively, MII oocytes can be obtained by in vitro maturation of GV oocytes. By denuding GV oocytes from preovulatory follicles, oocytes resume meiosis accompanied by GVBD and chromosome condensation. After insemination, each pronucleus is reformed with the decondensation of male and female chromatin. During the development of preimplantation embryos, endogenous PML nuclear bodies (PML-NBs) appear in the nucleus of blastomeres. Analysis of the exogenously formed PML-NBs in the nucleus of the GV oocyte is a model experiment to gain insight into the benefit of a NB-free intranuclear milieu in oocytes. (**B**) Representative z-stack image of exogenously formed PML-NBs (hPML VI -sfGFP mRNA-derived PML-NBs; hmdPML-NBs) in the nucleus of the GV oocyte. GV oocytes were injected with 25 ng/µL hPML VI -sfGFP mRNA and cultured for 48 h. BF, bright-field image of the oocyte. Scale bars, 19.9 µm.

**Figure 2 ijms-25-08656-f002:**
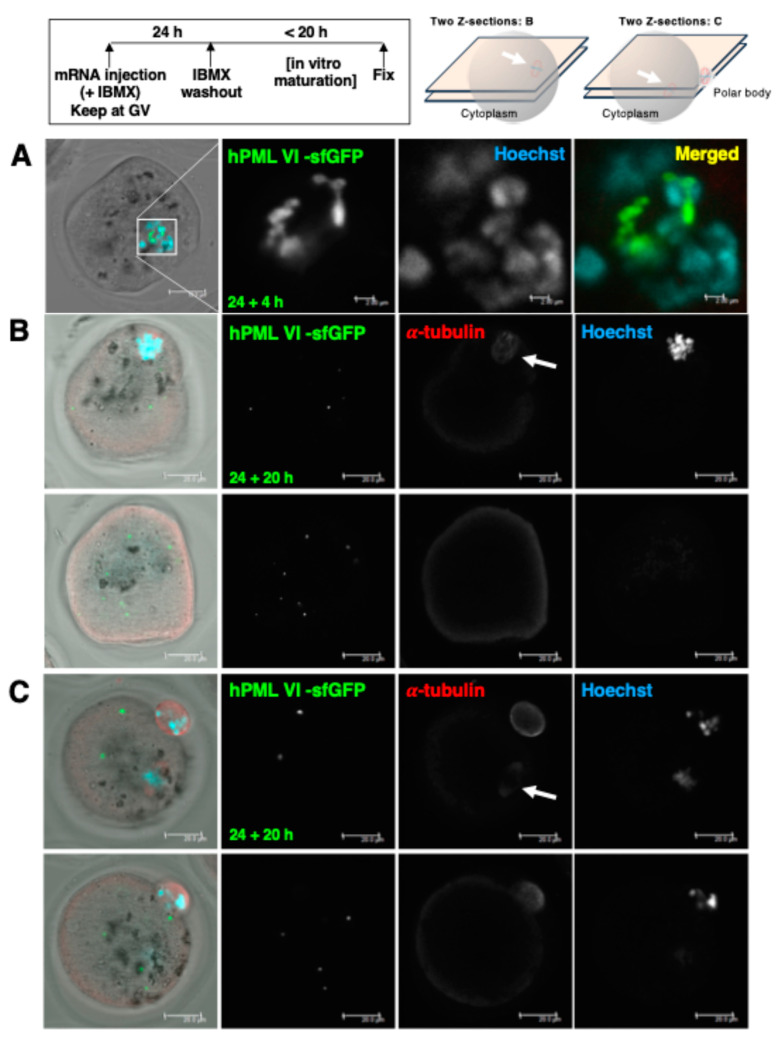
PML-NBs expelled from the nucleus upon GVBD stall in the cytoplasm. Representative images of preassembled hmdPML-NBs in the oocyte at (**A**) GVBD, (**B**) MI stage, and (**C**) MII stage. Oocytes undergo in vitro maturation for (**A**) 4 h or (**B**,**C**) 20 h. (**B**,**C**) Representative z-sectional images (as depicted at the top of the figure) that contain (upper panel, indicated by arrows) or do not contain (lower panel) α-tubulin-positive (red) spindles in the cytoplasm. Oocytes were stained with anti-human PML (green) and α-tubulin (red) antibodies. Scale bars, (**A**) 19.9 μm (for enlarged images of marked areas in the insets, 2.00 μm) or (**B**,**C**) 20.0 μm.

**Figure 3 ijms-25-08656-f003:**
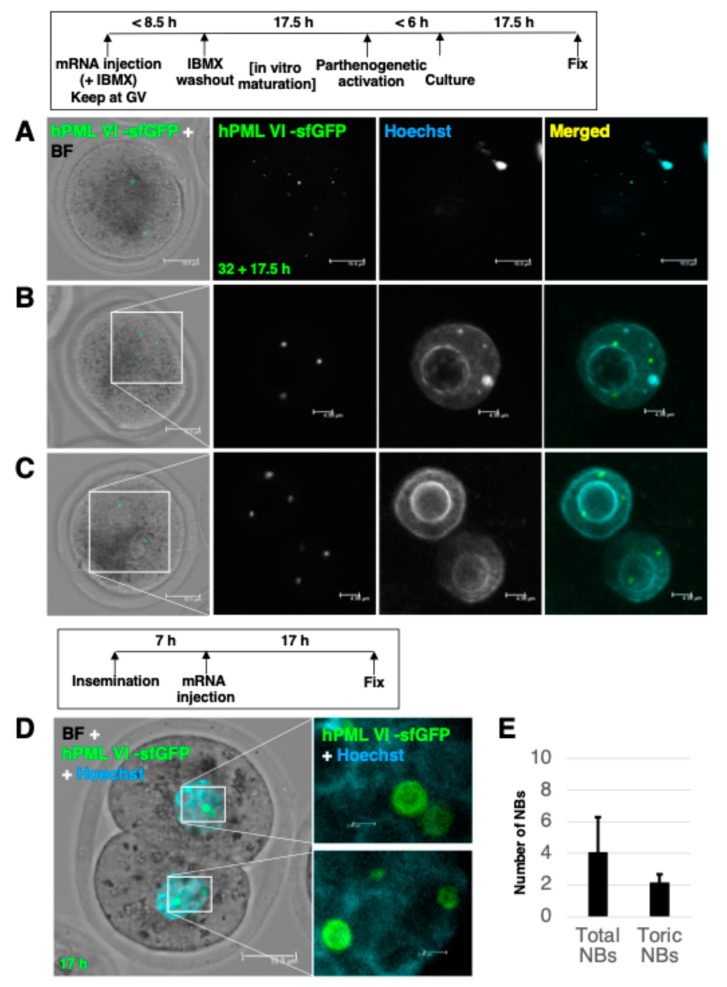
Stalled hmdPML-NBs in the cytoplasm are eliminated in activated zygotes with a pronucleus. (**A**, z-stack) Representative image of stalled hmdPML-NBs in the cytoplasm of oocytes arrested at MII stage. (**B**,**C**) Representative images of newly formed hmdPML-NBs in the pronucleus of zygotes. (**D**) Representative image of newly formed hmdPML-NBs in the nucleus of 2-cell embryo. Inseminated zygotes were injected with hPML VI -sfGFP mRNA and cultured for 17 h. (**E**) Quantification of the total/toric numbers of NBs formed in each pronucleus. Oocytes/zygotes were stained with anti-human PML (green) antibody. Scale bars, (**A**) 19.9 μm, (**B**–**D**) 19.9 μm. For enlarged images of marked areas in the insets, (**B**,**C**) 4.98 μm, (**D**) 2.00 μm.

**Figure 4 ijms-25-08656-f004:**
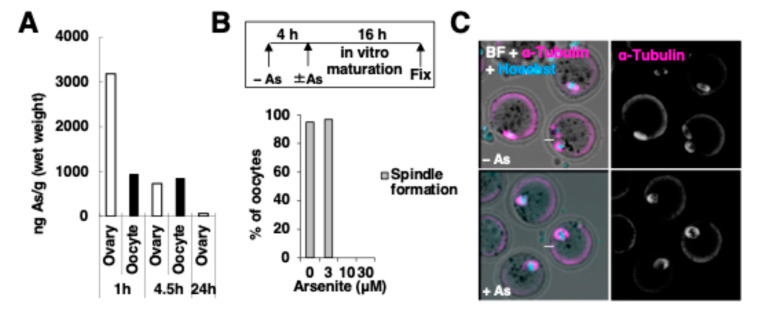
The stalemate of hmdPML-NBs in the cytoplasm is potentially exacerbated by stress. (**A**) Concentrations of arsenic in the ovary and oocytes measured by ICP-MS. Ovaries or oocytes were collected at indicated time courses after the intraperitoneal administration of arsenite. Shown data are average values from two independent ovarian samples or two independent sets (obtained from 7–9 mice) of oocyte samples. For the oocyte sample at 24 h, the concentration of arsenic was under the detection limit even with 566 oocytes. (**B**) Effect of arsenite on the spindle formation was scored in vitro. Oocytes were first cultured for 4 h in the arsenite-free condition followed by 16 h of culture in the presence or absence of arsenite. Number of oocytes examined, MII spindle and Polar body extrusion: 19–31 oocytes. (**C**) Representative images of the spindle morphology in oocytes treated with 3 μM arsenite or left untreated. Oocytes were stained with anti-α-Tubulin antibody. (**D**–**F**) hmdPML-NBs are assembled during in vitro maturation in the presence or absence of arsenite. (**D**) The numbers of NBs detected in the cytoplasm of oocytes are plotted, * *p* < 0.05. (**E**, z-stack) Representative images of the stalled hmdPML-NBs in the cytoplasm of metaphase oocytes at 21 h. (**F**) Representative images of the hmdPML-NBs in the nucleus of arrested GV oocytes at 21 h. Oocytes were stained with anti-human PML (green) antibody. Scale bars, (**C**) 20 μm, (**E**,**F**) 19.9 μm.

**Figure 5 ijms-25-08656-f005:**
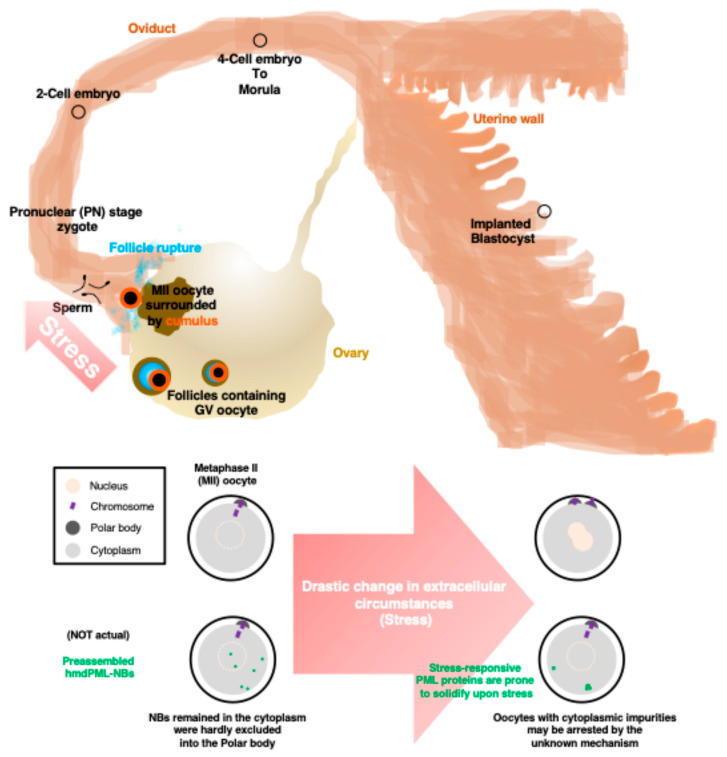
Schematic description of oocytes/embryos exposed to multiple stresses including the rupture from the ovary and the invasion of sperm. Artificially preassembled hmdPML-NBs persisted in the cytoplasm of oocytes. Since the stress-responsive property of PML protein would worsen the situation, the PML-NB-free intranuclear milieu of the oocyte would potentially have an advantage.

## Data Availability

Data is contained within the article and Appendix A.

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
