# Peer review of "Behavior of Assembled Promyelocytic Leukemia Nuclear Bodies upon Asymmetric Division in Mouse Oocytes"

_ijms, 2024, doi:10.3390/ijms25168656_

Round 1
Reviewer 1 Report
Comments and Suggestions for Authors
The author Osamu Udagawa, et al. have submitted a Manuscript (ijms-3075718) in International Journal of Molecular Sciences, entitled “PML-NBs-free nucleoplasmic environment is favorable for fertilization in mouse oocyte”.
The manuscript is a study conducted on promyelocytic leukemia (PML) nuclear bodies (PML-NBs) and here, the authors tested the assembly of PML-NBs into mice oocytes.
In consideration that PML-NBs is present in the nucleus of typical mammalian cell and is a multiprotein complex including small ubiquitin-like modifier (SUMO) that is been reported as one of the post-translational modifications of proteins in cells.
While in the mouse in the oocytes the PML-NB complex is disassembled in the cytoplasm and after fertilization during the developmental phases it was reassembled into new PML-NBs.
In this study, the authors focused on how PML-NBs can organize in the cytoplasm before rearranging in the pronucleus in fertilized oocytes.
These results likely suggest the performed analysis of culture of oocytes, zygotes, and embryos mice, treatment with RNA microinjection, expression plasmidico vector, immunofluorescent and others showing at the hand that authors have been obtained of PML-NBs reorganization in the nucleus of oocytes.
I believe that all data of this study may be useful to increase the knowledge on PML-NBs role in cells.
Finally, I consider the paper suitable for publication in International Journal of Molecular Sciences.
Reviewer 2 Report
Comments and Suggestions for Authors
Mammalian somatic cells, but not oocytes, have PML-NBs in the nuclei. The authors artificially induced the formation of PML-NBs in mouse oocytes to examine the consequence on fertilization.
1. The authors state that PML constitutes the core of the PML-NB that contains more than 100 proteins. It is not clear whether artificially induced PML-NBs also contain other proteins. The authors should comment on this, and discuss whether hPMLVI-sfGFP could self-aggregate into nuclear bodies.
2. Along the same line, the authors should discuss whether injection of synthetic mRNA encoding hPMLVI-sfGFP recapitulates the formation of PML-NBs under physiological condition.
3. The authors state that PML-NBs are unfavorable for fertilization. However, it is not very clear how the forced presence of PML-NBs impacts fertilization and early development. Do PML-NBs prevent maturation and fertilization? I cannot find related experiments in the manuscript.
4. In Figure, a-tubulin staining needs to be indicated.
5. Line 45, the statement “releasing the smaller blastomere as a first polar body” may be not appropriate. In general, a blastomere is the cell produced by cleavage divisions of the zygote.
Reviewer 3 Report
Comments and Suggestions for Authors
Article ID:ijms-3075718
Title: PPML-NBs-free nucleoplasmic environment is favorable for fertilization in mouse oocyte
Authors: Osamu Udagawa, Ayaka Kato-Udagawa, and Seishiro Hirano
In this manuscript, Udagawa and colleagues introduce Promyelocytic leukemia nuclear bodies in the murine ooxcyte (where they naturally don’t occur) and characterize the behavior of these nuclear bodies in the developing oocyte. Please see my comments below regarding this manuscript:
1. The authors need to spend some time making a case of why the mouse oocyte is a good model for this work. If there are no promyelocytic leukemia nuclear bodies in the mouse oocyte, then how do they know that the developmental processes which they observe in their study are not anomalous due to the alien environment where these nuclear bodies aren’t supposed to exist in the first place?
2. This is an interesting study of how PML bodies function in developing cells. However, I am not sure if they make a good case supporting their title of a PML-NBs-free nucleoplasmic environment being favorable to fertilization. Just because the oocyte removes the preassembled hmdPML-NBs doesn’t necessarily mean that it is favorable for fertilization. Maybe it is being removed for another reason?
In lines 50 - 52, they describe what could probably be a more accurate title:
3. The authors repeatedly use the term “exhaled” for the removal of PML-NBs. They may want to replace this term with “expelled”?
4. While it is obvious that Figure 2B and C are different, it is unclear what the reader should be looking at, and also why there are two rows. The authors should spend some time discussing the differences, especially in terms of the stained DNA, which is stained for but not discussed, except to say that hmdPML-NBs aren’t surrounding it?
5. Lines 222-223 The authors state “These results suggest that assembled hmdPML-NBs after fertilization do not show an overt defect on the progression of development.”.
Since these PML-NBs are in somatic cells of mice, are they supposed to be there under normal, control conditions? If so, why do the authors think that there could be a defect? Is this (the formation of new PML-NBs, the expression of KAP1 and DPPA2) part of the normal development? or is this unique to their experimental setup? It is unclear.
6. lines 247-248 “are promoted their degradation” should be changed to “which promotes their degradation”
7. Why did the authors choose arsenic for this study as toxin? Could it be that arsenite is interfering with the cellular processes? If not, the authors should state why this is not a possibility and that the results are strictly due to stress.
8. Lines 289-290 The authors state “Actually, they persisted in the cytoplasm of oocytes and became less-soluble debris unfavorable for fertilization”. How do they know it is unfavorable for fertilization? Did the authors show that fertilization was inhibited due to this debris in the cytoplasm? Did anyone else show this? If not, how can this statement be proven?
9. Lines 305 - 308 the authors state “As for the origin of newly formed hmdPML-NBs in the pronucleus, there was no difference in the number of NBs per pronucleus between the embryos with one and two pronuclei. Therefore, it was unlikely that the preassembled hmdPML-NBs in the cytoplasm were recycled or at least directly reused. “
This argument is not very strong. They could recycle and reassemble the same number of new hmdPML-NBs. No? Why would the number matter? Can the authors somewhere check to see if recycling is taking place? If not then this statement should not be made.
Round 2
Reviewer 2 Report
Comments and Suggestions for Authors
I have no further comments.